# Automatic Asymmetric Weight Distribution Detection and Correction Utilizing Electrical Muscle Stimulation

Kattoju Ravi Kiran*        Eugene Taranta†        Ryan Ghamandi‡        Joseph J.Laviola Jr.§

Interactive Systems and User Experience Lab
University of Central Florida, USA

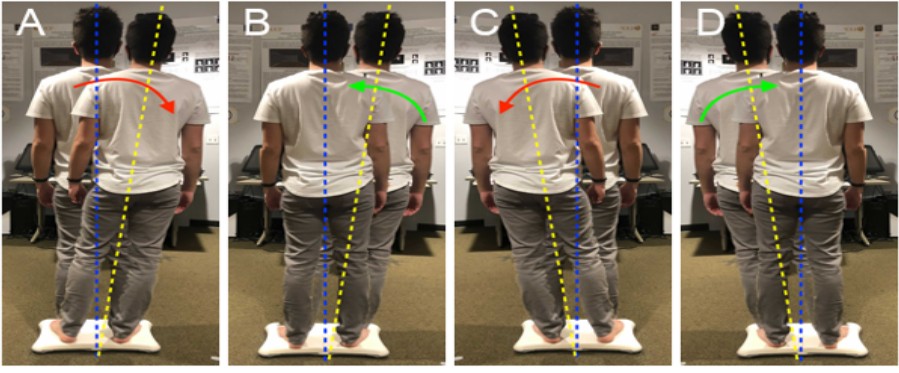

Figure 1: Impaired balance can have long-term health ramifications. Presented here are images of asymmetric weight distribution (AWD) due to prolonged standing and restored balance conditions using electrical muscle stimulation (EMS): (A) AWD right, (C) AWD left, (B) & (D) EMS feedback based stabilization and restoration of balanced posture. The red arrows indicate direction of progressive AWD and green arrows indicate a counter-weight shift balance stabilization due to EMS feedback correction to the tibialis muscle.

## ABSTRACT

Postural control is a constant re-establishment process for the maintenance of balance and stability. Asymmetric weight distribution (AWD), characterized by uneven leg loading, leads to increased instability, injury, and progressive deterioration of posture and gait. Postural self-correction is automatically affected by the human body in response to visual, vestibular, and proprioceptive sensory information. However, simultaneous cognitive loads can increase the demand for extra resources and require balance monitoring and correction techniques. We address these issues with a novel physiological feedback system that utilizes load sensors for AWD detection, and electrical muscle stimulation (EMS) for automatic correction and restoration of balance by affecting a counter-weight shift. In a user study involving 36 participants, we compare our automatic approach against two alternative feedback systems (Audio and Vibro-tactile). We find that our automatic approach delivered faster correction and outperformed alternative feedback mechanisms and perceived to be interesting, comfortable and a potential commercial product.

**Index Terms:** Human-centered computing—Human computer interaction (HCI)—Wearable computing—Preventive Healthcare Posture correction—Asymmetric weight distribution—Electrical Muscle Stimulation;

---

*e-mail:Kattoju.ravikiran@knights.ucf.edu
†e-mail:etaranta@gmail.com
‡e-mail:ryanghamandi1@gmail.com
§e-mail:jjl@cs.ucf.edu

## 1 INTRODUCTION

The maintenance of stable posture is important as two-thirds of our body mass, and delicate organs are being supported by our legs which form a narrow base of support. Asymmetric weight distribution (AWD) characterized by postural sway and impaired standing balance has been known to be responsible for multiple health conditions resulting in reduced functional ability [109]. Numerous posture-related health issues such as lower back pain [81], anterior cruciate ligament ruptures [46, 78, 86], and knee and ankle injuries [32, 64] are associated with an increase in postural sway and AWD. Postural control is a constant re-establishment process of balance and is integral to the safe execution of most movements in our daily life. Posture adjustment relies primarily on the integration of different sensory feedback such as the visual, vestibular, and proprioceptive control systems. Subconscious proprioception, in the form of awareness from muscle receptors, and joints also play an important role in the control of posture and balance. However, the effectiveness of our body's postural control system decreases with cognitive demand, age, and injuries, and imposes a critical demand on the postural control system especially while being engaged in additional cognitive tasks during standing activities. Although conscious proprioception plays a crucial role in gross muscular and full-body posture adjustments, poor postural habits and impaired proprioception may lead to increased postural sway, AWD, and even loss of balance [3]. AWD may lead to increasing instability, subsequent injury, and progressive deterioration of posture and gait [107]. Investigation of AWD has provided valuable information in an array of situations such as fall detection and prediction in the elderly [48], evaluation of balance-related disabilities (Parkinson's disease, stroke, and concussions), and lower body post-surgery rehabilitation [1, 2, 30, 79, 99].

Nearly $90 billion are spent annually in the USA, for treating repetitive strain injuries (RSI) and lower body injuries arising out of poor workplace postures and prolonged standing [20, 22]. Prolonged

standing causes muscle dysfunction, or dystrophy of the muscles of the leg and often leads to unequal load distribution on the hips, knees, ankles, and feet which are responsible for stabilizing the torso in an upright position and is directly associated with lower back pain [94]. Lower body injuries are one of the noted root causes of disability in the world and affect approximately 80% of the world population at some point in their lives [54, 98]. As existing intervention technology attempts only postural sway detection and necessitates the participants' attention and effort to self-correct imbalance, there is a need for the development of an automatic wearable intervention technology with the capability for AWD detection and subsequent correction to facilitate proper posture maintenance during tasks involving prolonged standing hours such as work, recreational, and gaming activities.

As EMS has been shown to induce involuntary muscular contractions for generating physiological responses [21, 91, 102], we integrated EMS with an AWD detection system to automatically detect and correct habitual AWD posture and restore balance in posture through involuntary contractions of the muscles in the legs. Our work aims to explore and provide insights into differences between our approach of automatic posture correction and self-correction in traditional feedback techniques. We evaluated the performance of our automatic approach across two different applications with varying levels of engagement and posture awareness, and a novel between-subjects study was conducted. The performance of our automatic approach was measured by the correction response times to the EMS feedback. Qualitative data in the form of user perception rankings for different usability parameters were recorded and analyzed. In comparison to the previous research, the main contributions of this work include

1. The development of a novel intervention prototype that autonomously detects and corrects AWD posture through a physiological feedback loop utilizing EMS.

2. A user study for quantitative, and qualitative evaluation of performance, and usability of our automatic AWD detection and correction utilizing EMS feedback against two traditional feedback techniques (audio and vibro-tactile), and under two different conditions of posture awareness and engagement in breaking the habit of AWD, and for training and developing good postural habits.

## 2 RELATED WORK

Owing to the increasing awareness of workplace injuries, health, and wellness, there has been a renewed interest in the relationship between postural control and cognitive load in recent times [3]. Self-correction of posture is affected automatically by the human body to a certain extent in response to sensory information such as visual, vestibular, and proprioceptive information. However, any additional loads due to simultaneous cognitive tasks demand extra resources, and necessitates balance monitoring and correction techniques [4, 55, 62, 80, 105]. Previous research on AWD monitoring and detection can be classified into two main categories: balance and stability monitoring, and asymmetric weight distribution with real-time feedback solutions.

### 2.1 Balance and Stability Monitoring

Balance and stability monitoring has primarily been an area of research for detecting neurological disorders, gait imbalance, lower-body injury, and post-surgery rehabilitation. Traditionally, the measurement of impaired balance and AWD employed highly specialized equipment such as force plates [6, 43], electrogoniometers [87], video motion analysis [23], electromyography [82], and magnetic tracking systems [101]. Balance and stability monitoring techniques using force plates often measured the center

of pressure/gravity and balance ratios [35], while Inertial measurement units (IMU) [7, 11, 36, 90] and video analysis techniques [45, 47, 50, 113] relied on computed angular changes. However, expensive equipment developed for medical rehabilitation and clinical research was found to be cumbersome due to the attachment of markers and sensors to the skin/clothing. This resulted in difficulties in conducting easy, non-invasive data collection concerning AWD. As a cost-effective alternative, standing balance has also been evaluated using a Wii Balance Board (WBB) in different clinical settings [16, 17, 34, 111]. WBB was utilized in clinical trials in brain injury patients to determine the effectiveness of balance rehabilitation [34], and predicting fall risks in older adults [111]. Additionally, other researchers also investigated postural sway and standing balance in a quiet standing condition among young adults [5, 66, 89], elderly [8, 25, 26], athletes [61, 65], and brain injury, and Parkinson's Disease (PD) patients [31, 97, 106].

Research on postural sway was conducted to investigate steadiness in different stances [35] and different postural control tasks [17], to determine influence of standing duration on sway [63], and to expose impairments leading to disequilibrium and evaluate compensatory strategies in quiet standing positions in patients [9]. Postural sway was also investigated to determine the effect of dual tasks on standing balance [92]. Researchers also developed postural control strategies for clinical rehabilitation of patients suffering from Parkinson's disease (PD), and diagnosing sports-based impairments by investigating effects of altered postural control and balance on the ankle and hip in PD patients during quiet standing [7].Further, researchers also investigated the effects of anticipatory postural adjustments in patients with PD [11], detecting balance irregularities in athletes at risk of AWD [90], incidence of head impacts due to imbalance [36], and sway assessment for detecting balance impairments in athletic populations. Finally, postural sway and balance impairment studies have also been conducted by different researchers for postural control in concussion patients [24], neurological disorders [112], and injury prevention [95]. However, the above-mentioned research studies are only focused on the assessment and monitoring of balance and postural sway for diagnosis of balance impairments, and the development of rehabilitation protocols for balance training. These techniques do not provide any posture correction feedback to the participants. To address this, our research focuses on both detection of AWD conditions and subsequently provide real time automatic correction feedback to restore balance using EMS.
.

### 2.2 Asymmetric Weight Distribution Detection with Feedback

Maintaining balance and stability is a complex activity that is accomplished by a synergy between the brain and different sensory information from the vestibular, somatosensory, and visual systems. Postural instability or abnormal postural sway coincides with asymmetric weight distribution or weight-bearing asymmetry when feedback from sensory systems is inaccurate. However, this loss or absence of sensory information can be compensated by providing additional external sensory feedback to the brain for effecting posture correction and maintaining balance [44, 83]. Due to the advancements in sensor technology, and smarter algorithms, the past decade has seen an increased interest in the design and development of biofeedback-based postural control devices for maintaining balance.

Audio feedback systems were developed by researchers for improving balance in patients suffering from bilateral vestibular loss [27], for comparing the effect of visual senses and environmental conditions on postural control [28], and for improving balance in comparison to absence or unreliable sensory feedback [14]. Alternatively, visual feedback was utilized to evaluate the effectiveness of human balance improvement in quiet standing tasks [39], to explore the effect of interactive balance training on postural stability in daily

physical activities [37], and for developing balance rehabilitation strategies based on ankle movement to compensate for impaired joint proprioception in patients [38]. Augmented sensory feedback through visual, and auditory feedback was further explored to investigate the relative effectiveness of the two types of feedback on improving postural control [41] and it was concluded that audio feedback was more effective for motor learning and maintaining balance. Virtual reality integrated with visual feedback has also been employed in developing balance training rehabilitation protocols and biofeedback for minimizing fall risks [108], to investigate the influence of moving visual immersive environments on postural control [56], improving standing balance in patients suffering from hemiplegia [10], and PD [33]. All the above-mentioned balance and stability detection techniques focused on alerting the user through traditional audio, visual or vibro-tactile feedback techniques and relied entirely on the participants' ability to process the feedback, and their willingness to self-correct their AWD. Although, these AWD detection techniques enabled minimization of postural sway and restoration of balance using different types of feedback, they still required the users' willingness to self-correct posture when AWD or postural sway is detected. Additionally no posture correction feedback response times and user perception parameters have been reported. The traditional feedback types are also known to place a cognitive load on the user by relying solely on the user's intent and desire to self-correct their posture based on the received feedback especially when engaged in a cognitively demanding task [44, 83].

Additionally, clinical research on gait rehabilitation for stroke patients has been conducted through the application of electrical stimulation to the gluteus medius and tibialis anterior [58], hip abductor and ankle dorsiflexor [15]. However, these systems are not automatic and utilize a manual trigger mechanism for providing the correction feedback to improve spatio-temporal parameters during dynamic activities like walking by controlling pronation, and foot placement during walking activities. Further, the tactile component of EMS was utilized on the thigh to help provide notifications to improve walking gait in post-stroke survivors [57]. Their system utilized force sensitive resistors capable of detecting impact of heel and foot strikes for detecting improper gait during walking activities and utilized EMS to provide only a vibro-tactile sensory stimulation without invoking any involuntary muscular activity that can alter the patient's gait. Their technique also relied on the user to make a conscious effort to correct their pronation and foot strikes to improve gait. Although, the above techniques utilized EMS for gait rehabilitation in clinical settings to address pronation, foot placement, and foot striking in dynamic activities such as walking in stroke patients, AWD detection and subsequent automatic balance stabilization during prolonged standing conditions in every day activities is not fully explored. This presents a gap in the research for the design and development of autonomous AWD detection and correction systems for preventing fall risks, gait imbalance, and proper rehabilitation after injury and surgery. Our automatic AWD detection and correction prototype system addresses this research gap by employing EMS to automatically generate a physiological counter weight shift response through involuntary muscular contractions of the tibialis muscle for restoring balanced posture when AWD conditions are detected during two prolonged standing conditions (quiet Standing and mobile gaming) under different levels of posture awareness, thereby reducing the additional cognitive load required for self correcting their posture.

## 2.3 Electrical Muscle Stimulation (EMS)

Primarily, EMS has been utilized in pain management therapy to deliver electrical impulses to the muscles, nerves, and joints in a non-invasive manner via surface electrodes placed on the skin. Besides being used for alleviating chronic conditions of muscle strains and spasms, EMS was also employed in post-surgery rehabilitation to regain normal function [102], and in post-injury recovery for rebuilding muscle strength [12, 21]. EMS has also been applied in clinical research to generate involuntary muscle contractions for restoring normal function to impaired muscles due to injury, surgery, or disuse, and also to restore normal functional actions such as hand grasping in hemiplegic patients [21], generating reflex actions for disorders involving swallowing [96], and to enable control of neuro-prosthetic implants [91].

## 2.4 EMS in Human Computer Interaction

The capability of EMS to deliver haptic and somatosensory feedback has led to a newfound interest in the human-computer interaction (HCI) domain for the development of immersive training, and gaming in virtual, augmented, and mixed reality applications [59, 67–70, 100]. Due to its adaptability, EMS has enabled the development of new interactive approaches for dynamic activity training, delivering more immersive experiences through somatosensory feedback, and in the development of spatial interfaces for user interaction. Dynamic activity training using EMS has been explored to enable users to acquire and develop new motor skills such as learning to play a musical instrument [104], learn offered affordances of different objects [74], and enable the development of fast reflexes for preemptive actions [51, 52, 84]. Additionally, with its ability to generate physiological responses through invoked involuntary muscular contractions, EMS had been utilized to develop force feedback applications to emulate impact [29, 71], increase dexterity by flexing individual fingers [103], apply physical forces to gaming devices [77], objects [76], and walls and barriers in virtual environments [72, 75]. EMS has also permitted researchers to develop increased immersion in virtual reality applications through sharing kinesthetic experiences from tremors in patients with Parkinson's disease [85], arousing fear and pain in *In-pulse* [60], and transmitting emotions between individuals in *Emotion Actuator* [42]. Further, integration of EMS with input/output devices has enabled the development of physiological feedback loops in *Pose-IO* for proprioceptive interaction [73], induced navigation [88], bio-metric user authentication [13], influenced sketching [77], running assistant [19], discrete notification systems [40], and involuntary motor learning [18].

The current literature suggests that traditional feedback-based posture alert systems relied entirely on the users' intent and willingness to correct their improper posture and that EMS feedback-based posture correction has not been fully explored. Although, the above-mentioned characteristic interactive and adaptive features of EMS-based technologies have validated its ability to deliver latent, distinct, and more distinguished feedback for delivering immersive experiences, dynamic activity training, and input/output interfaces, our work investigates the feasibility of automatic posture correction for restoring balance and stabilization through a counter weight shift strategy utilizing EMS.

## 3 AUTOMATIC DETECTION AND CORRECTION OF AWD

For automatic detection and correction of AWD, we developed an intervention prototype based on a physiological feedback loop that relied on load sensors and EMS (illustrated in Figure 2). Our prototype employed a wireless Wii Balance Board (WBB) for measuring changes in weight distribution across the two legs using the balance ratio of the weights displaced by the two legs separately, and the openEMSstim package [69] for presenting the EMS correction feedback. A C#-based user interface using a Wii-mote library was developed to integrate the WBB with the EMS hardware to complete the physiological feedback loop. As AWD is mainly characterized by progressive and/or unusual leaning to either side [49], our system was designed to detect these changes in weight distribution across the two legs using the shift in balance ratio representing the AWD conditions.

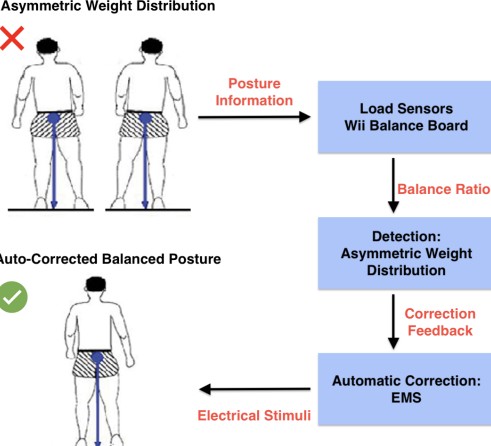

Figure 2: Physiological feedback loop: Automatic asymmetric weight distribution detection and correction system. Asymmetric weight distribution posture (top) illustrates leaning to either side and the auto-corrected posture (bottom) illustrates the restored balanced posture through counter weight shift using EMS.

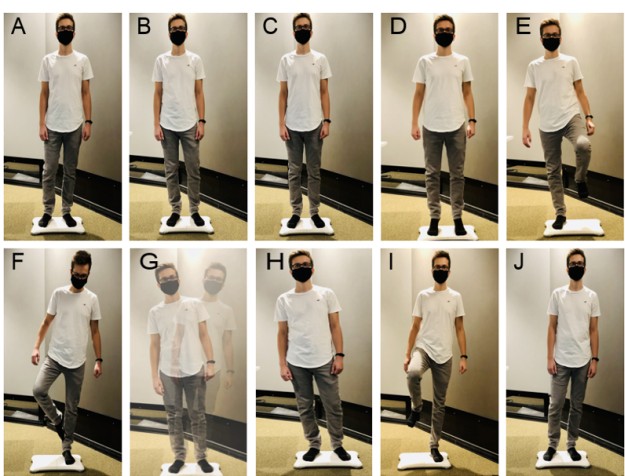

Figure 3: Some examples of typical actions performed during standing activities based on movement observations of employees taking breaks after standing. (A) Lean slight left, (B) Lean slight right, (C) Balanced, (D) Calf raise and reset, (E) Lift left leg and reset, (F) Scratch leg and reset, (G) Sway and reset, (H) Lean extreme right, (I) Lift right leg and reset, (J) Lean extreme left.

## 3.1 Time and Balance Thresholds

Asymmetrical leg loading can be detected from the shift in balance ratio calculated from the weight displacement information obtained from the load sensors in the WBB. Our proposed system detected AWD when the user's balance ratio approached and crossed preset balance ratio and time thresholds. To improve our system robustness and tune our system for optimal performance, we collected ecologically valid balance ratio data from 10 participants performing 10 typical actions one performs consciously or unconsciously when they are standing idly (illustrated in Figure 3). These 10 unique actions were identified based on general movement observations of employees taking breaks from standing. These actions were interleaved with moderate and extreme leaning actions to ensure AWD conditions were embedded in each session. The balance ratio patterns of the 10 actions are shown in Figure 4. A grid search was then employed to find the balance ratio and time thresholds that optimized the accuracy of AWD detection. Since our primary concern was the impact of false positives on user perception and to prevent unwarranted correction feedback, we selected thresholds that minimized false positives first, maximized true positives second, and maximized the per-frame Jaccard index of similarity [93] with the manually marked per-frame ground truth third. With valid data collected from 10 participants, using a leave-one-subject-out protocol, we found that at a time threshold of 2.9 seconds and balance ratio threshold of 1.25, our system achieved high accuracy of 96% for true positive AWD detection, 0.1% for false-positive AWD detection, and 0.3% for false rate. The balance ratio of 1.25 translates to a left-to-right or right-to-left AWD balance ratio of 55.5 : 44.5.

The preset time and balance ratio thresholds obtained through our tuning process allowed the AWD detection system to overcome measurement errors, mitigate false positives, and ensured that typical movements such as actions illustrated in Figure 3 did not lead to false-positive AWD detection or activate unwarranted correction feedback. When the user's balance ratio approached and crossed the preset balance ratio threshold of 1.25, a countdown timer set to the preset time threshold value of 2.9 seconds was initiated to provide correction feedback after the time threshold had elapsed. The purpose of the timer is to ensure that false positives due to participant behavior do not trigger a correction feedback response.

## 3.2 Correction Feedback

The Wii balance board contains load sensors at each corner (top left, bottom left, top right, and bottom right) allowing measurement of the weight distributed across each leg and calculation of the weight balance ratio for AWD detection. When AWD is detected, automatic correction feedback would be presented to the user by applying electrical stimulus to the tibialis muscles for generating a counter-weight shift force in the opposite leg to the direction of the AWD leaning and thereby, generating a physiological response to stabilize the user back to a 50:50 balanced equal weight distribution position. A pair of electrodes on each leg (illustrated in Figure 5) would be utilized for contraction of the tibialis muscle which causes the foot to roll outward, thus generating a physiological response of a counter-weight shift. This generated counter-weight shift attempts to redistribute the weight more evenly across the two legs, thereby stabilizing the user back to the balanced 50:50 weight distribution position. Calibration of the WBB and EMS intensity play a crucial role in the effectiveness of the system. The calibration process includes correcting offset values of the load sensors in the WBB prior to start of the study session. The users' balance ratio in balanced position and emulated AWD leaning positions relative to the balanced position are monitored to ensure WBB is calibrated. For the EMS calibration, the EMS intensity would be manually incremented to deliver an intensity that is optimal for generating involuntary muscular contraction, comfortable, and avoid any discomfort or pain to the user. This EMS intensity, provided to the user for generating the necessary force for correcting AWD posture and restoring the balanced position, would be recorded and utilized during the experiment. The Trans-cutaneous electrical stimulation ($TENS$) device can deliver intensities between (0-70 $mA$). A continuous square wave at a pulse width of 100 $\mu$s with a frequency of 75 $Hz$ at the recorded EMS intensity would be presented as EMS feedback to the participants. The EMS calibration procedure is described in detail in section 4.5.

## 3.3 Operation

Our Physiological feedback loop for detecting and correcting AWD relied on the changes in balance ratio along with the total weight distributed on each leg. This allowed our system to detect AWD left/right conditions when the balance and time thresholds have been crossed. AWD occurs when a user unevenly distributed body

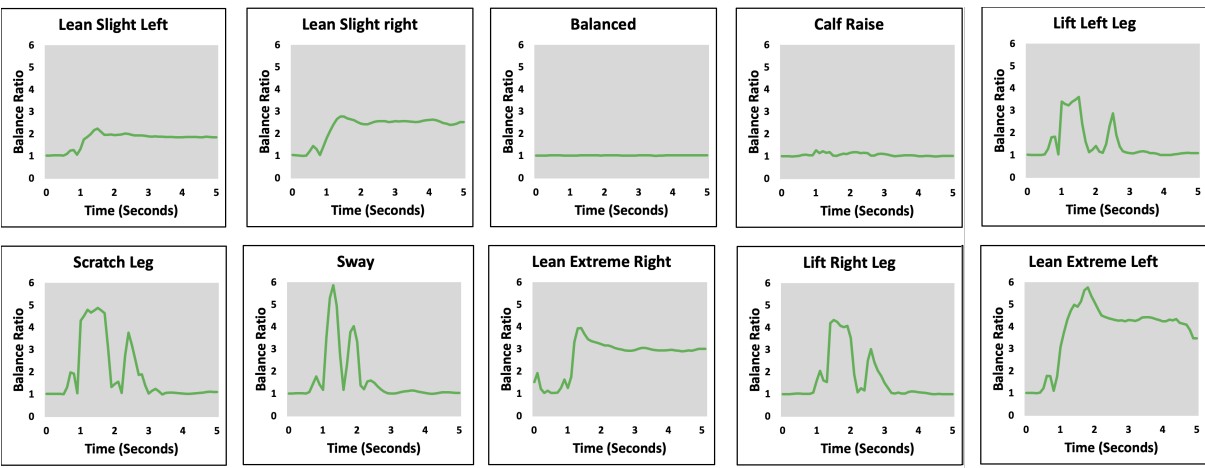

Figure 4: Balance ratio patterns of the 10 actions performed by users (illustrated in Figure 3) for the tuning process to determine balance and time thresholds for AWD detection. The lean actions representative of AWD exhibited higher balance ratios and for prolonged time durations in comparison to the other actions.

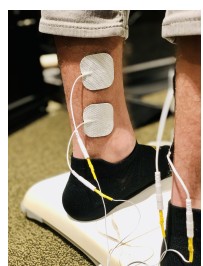

Figure 5: EMS electrode placement on tibialis muscle for affecting counter weight shift.

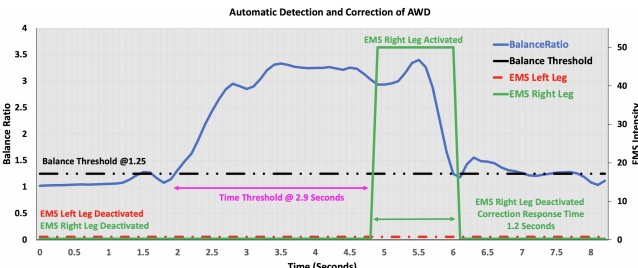

Figure 6: Automatic detection and correction of AWD: Graph showing EMS activation and deactivation. When the user's balance ratio approached and crossed preset balance ratio and time thresholds, EMS was activated for AWD correction. EMS was deactivated when 50:50 balance was restored.

weight across the two legs. This places an additional stress on the ankle, knee, hip, and lower back. To detect these AWD conditions, our system utilized the balance and time thresholds determined in Section 3.1. Figure 6 illustrates the activation and deactivation of EMS correction feedback when an AWD left condition was detected and corrected for a participant during the study. Initially, under a balanced posture condition, the EMS left leg and EMS right leg remain deactivated. A timer with preset time threshold of 2.9 *Seconds* was activated when the user's balance ratio gradually increased and crossed the preset threshold of 1.2. Upon completion of the timer, if the balance ratio still remained above the threshold, the EMS was activated to apply a stimulus of 50 *mA* to invoke a muscular contraction on the right tibialis muscle (EMS Right Leg) for generating a counter weight shift and restoring balanced posture. The EMS was deactivated immediately after the balanced posture is restored. A correction response time of 1.2 *Seconds* was recorded between activation and deactivation of the EMS Right Leg. The AWD right condition is similarly detected and corrected by activating and deactivating the EMS Left Leg.

## 4 METHODS

The goal of this study was to evaluate the overall effectiveness and user perception of our automatic AWD detection and correction feedback system using EMS compared to traditional audio and vibro-tactile feedback modalities. The audio and vibro-tactile feedback modalities required self-correction by the user based on audio and vibro-tactile notifications delivered to them, respectively. We also identified two common use cases of everyday activities with varying levels of engagement and posture awareness such as quiet standing (QS) and playing a mobile game (MG) (illustrated in Figure 7) to in-

vestigate the effect of cognitive demand on posture awareness, AWD occurrence, and type of correction feedback. Our objective was to determine if our automatic AWD detection and correction system using EMS feedback would be a viable technique for correcting AWD as opposed to the audio and the vibro-tactile feedback types while standing idly or being engaged in cognitively demanding task.

### 4.1 Subjects and Apparatus

We recruited 36 participants ($Male = 29$, $Female = 7$) for the study with 18 participants for each application-quiet standing, and mobile game. All participants were aged 18 *years* and above with mean age of 24.67 *years* ($S.D. = 3.98$ *years*), mean weight of 71.1 $Kg$ ($S.D = 10.88$ $Kg$), and mean height of 167.3 $cm$ ($S.D = 8.94$ $cm$). All participants were able-bodied and had corrective 20/20 vision. For monitoring the balance ratio along the medial lateral axis, a Wii balance board was utilized. A Grove-vibration motor with double-sided disposable adhesives was utilized for delivering the vibro-tactile feedback (illustrated in Figure 9 (a)). An off-the-shelf TENS unit ($TN$ $SM$ $MF2$), and $openEMSStim$ package [68] was utilized for generating the EMS feedback and controlling the activation and modulation of the intensity of the electrical stimuli supplied to the muscles, respectively. A 14" Intel $i7$ laptop was utilized for the study user interface and an iPhone $SE$ $2nd$ generation was employed for the mobile game application. Qualitative data from the pre-questionnaire survey on participants' prior exposure to balance alert devices and EMS, experience with posture problems, and AWD is illustrated in Table 1. Participants ranked their exposure

Table 1: User ranking on posture awareness, devices, and EMS. User ranking on a 7-point Likert scale. QS: Quiet standing, MG: Mobile game

| User Experience | Application | Mean | S.D |
|---|---|---|---|
| Exposure to balance alert devices | QS | 1.44 | 0.70 |
| | MG | 2.11 | 1.28 |
| Exposure to EMS | QS | 2.56 | 1.39 |
| | MG | 1.94 | 1.25 |
| Prolonged standing | QS | 4.39 | 1.87 |
| | MG | 4.11 | 1.67 |
| Experienced AWD | QS | 4.33 | 2.01 |
| | MG | 3.67 | 2.08 |

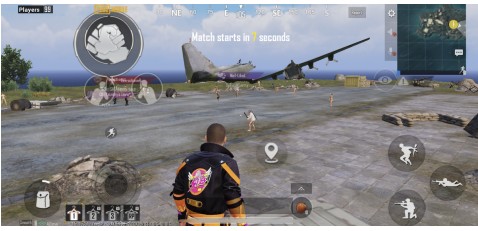

Figure 7: Participants played PUBG mobile in the mobile game condition. Image shows the lobby area of the game prior to starting.

and experience on a 7-point Likert scale with 1 meaning never/no experience and 7 meaning frequently/very experienced.

### 4.2 Experimental Design

To investigate the performance and feasibility of our approach, a 2 by 3 mixed subjects experiment with 36 participants was conducted. The within-subject factor was the feedback type (audio, vibro-tactile, and EMS) and the between subject factor was the application type (Quiet standing (QS) and Mobile game (MG)). The performance of our automatic AWD correction using the EMS feedback was compared against the self-correction in the audio and vibro-tactile feedback techniques. A quantitative evaluation of the average correction response times and a qualitative evaluation of the perceived usability of our system was conducted across the three feedback and the two application types. In both applications, participants were required to stand on the WBB without shoes for three 15-minute sessions, one for each of the three modalities listed below. In the quiet standing application, participants were required to stand quietly (illustrated in Figure 8 (A), (B), & (C)), while participants played a mobile version of "PlayerUnknown's Battlegrounds (PUBG)"[1] in the mobile game application (illustrated in Figure 8 (D), (E), & (F)). PUBG mobile is an engaging battle royale game (illustrated in Figure 7) and was selected for this study due to its high engagement level and popularity amongst people aged between $15-35$ years, who may be more prone to AWD due to prolonged standing hours at work or mobile gaming sessions. In both applications, participants were required to complete the following three modalities:

- **Modality 1**: Audio alert feedback and self-correction
- **Modality 2**: Vibro-tactile alert feedback and self-correction
- **Modality 3**: EMS feedback and automatic correction

In both applications, the order in which the participants were introduced to the modalities was counterbalanced to minimize learning

---

[1]https://www.pubg.com/

effects. The three different modalities and the two applications in the study were the independent variables and the dependent variables were the average correction response times, and user perception parameters such as accuracy of correction feedback, task disruption, comfort, and posture awareness. Each study session lasted approximately $60-75$ minutes and the participants were compensated $15 for their participation.

### 4.3 Research Hypotheses

Our study was designed to determine the effects of automatic or self-posture correction on user experience across the two applications, and three feedback modalities. As such, we expect to find the main or interaction effects of modality and application type on the average correction response times, and the user perception of correction feedback accuracy, comfort, disruption. EMS being an semi-invasive feedback technology, we developed four research hypotheses below to determine the usability of EMS for AWD correction against the traditional audio and vibrotactile feedbacks.

- **H1:** Average correction response times to EMS feedback will be the fastest among all three modalities.

- **H2:** Correction feedback accuracy in the EMS feedback modality will be greater in comparison to the other modalities.

- **H3:** EMS feedback modality will be equally comfortable as the alternative traditional feedback types and across both application types.

- **H4:** EMS feedback modality will be more disruptive across the three modalities.

### 4.4 COVID-19 Considerations

Due to the ongoing COVID-19 pandemic, we wanted to ensure safety for the participants and researchers. Following our institution's guidelines, all individuals were required to always wear face masks. Between each user, we sanitized all devices and surfaces that the participants and researchers would be in contact with. We also provided hand sanitizer, cleaning wipes, and latex gloves to reduce the risk of contracting the disease.

### 4.5 Experimental Procedures

Before the start of the study session, participants were required to review the consent document and provide their consent for participating in the research. Participants then completed a pre-questionnaire survey on knowledge and experience with balance-related intervention technology, AWD, and EMS. Upon completion of the pre-questionnaire survey, participants were required to complete a validation study where they performed a set of the 10 typical actions on the WBB as illustrated in the Figure 3 to ensure the AWD detection system with the preset balance threshold (1.25) and time threshold (2.9 seconds) was able to detect the AWD conditions (Lean slight right/left, Lean extreme right/left) accurately and to mitigate the possibility of false-positive correction feedback. Next, participants were required to stand without shoes on the WBB for calibration. For the vibro-tactile alert modality, Grove vibration motors were placed on each leg with double-sided adhesives as illustrated in Figure 9 (a). Adhesive EMS electrodes were placed on each leg along the tibialis muscles before the EMS feedback session for correcting AWD as illustrated in Figure 9 (b). Before the EMS feedback session, participants were required to stand on the WBB and were calibrated for an optimal EMS intensity that affected balance stabilization and corrected AWD posture. Each user's optimal EMS intensity level was manually calibrated by the study moderator only once. Participants were asked to emulate an AWD condition of leaning left or right and the moderators incremented the EMS intensity on the opposite leg until an involuntary muscular contraction is felt by the

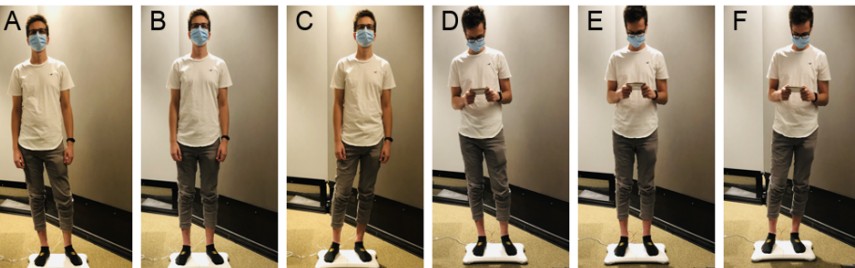

Figure 8: Evaluation of the effectiveness of our automatic approach across 2 different application types- Quiet Standing (A), (B), (C) and Mobile Game (D), (E), (F). Quiet Standing: (A) AWD right, (B) Balanced, (C) AWD left. Mobile Game: (D) AWD right, (E) Balanced, (F) AWD left.

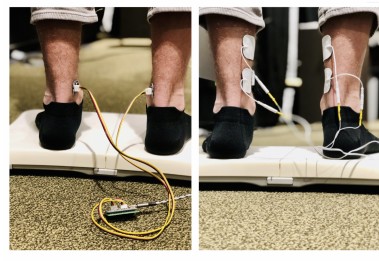

(a) Haptic motor unit placement    (b) EMS electrode placement

Figure 9: Haptic motor unit and EMS electrode placement on the tibialis muscle. (a) Vibro-tactile feedback is delivered to the legs through the haptic motor units placed on each leg. (b) EMS feedback is delivered through EMS Electrodes place on the tibialis muscle on each leg.

user and generated a physiological response of a counter-weight shift in an attempt to stabilize the balance ratio. The above process was repeated for both AWD left and AWD right conditions to deliver the user with an optimal user experience in the EMS feedback session. As EMS has been known to produce a haptic effect at low intensities, participants were asked to ignore the haptic effect to ensure the haptic component did not contribute to the automatic AWD correction process in any way. Additionally, during this calibration process, moderators also asked participants to specifically respond verbally to the following questions to ensure tibialis muscular contraction and user comfort: 1) If and when they initially felt a haptic sensation of the EMS, 2) If and when they felt the EMS intensity generating an involuntary contraction in the leg and/or when they are experiencing the counter-weight shift force towards restoring their balance, 3) If and when they felt any pain or discomfort. For each user, this involuntary muscular contraction affecting AWD correction was visually verified by the moderator and verbally confirmed by the user. The optimal EMS intensity which generated the counter-weight shift effect to correct AWD and was also comfortable to the user was recorded to be used for the EMS feedback session of the study.

The above EMS intensity calibration steps are similar in both the quiet standing and the mobile game applications. In the quiet standing application, participants would be asked to stand quietly, while for the mobile game application, participants would be required to play PUBG. In both applications, participants would be required to stand without shoes on the WBB, and their balance ratio would be monitored for AWD (illustrated in Figure 8). The study comprises three parts: audio, vibro-tactile, and EMS feedback. Each part of the study is 15 *minutes* in duration and all participants were required to finish all three parts to complete the study. The participants were given a 5-*minute* seated break after each part of the study, where participants were required to remain seated to rest their legs. Participants then completed a survey about their experience after each part.

### 4.5.1 Audio feedback and self-correction:

Upon AWD detection based on balance ratio from the WBB, an audio notification "*Leaning left/right-please correct imbalance*" is activated and the participants were required to self-correct their AWD posture and stabilize their balance till another audio notification "*Stabilized*" is presented to them.

### 4.5.2 Vibro-tactile feedback and self-correction:

Upon AWD detection based on balance ratio from the WBB, a vibro-tactile notification in the form of vibration from the haptic motor is activated on the opposite leg, indicating the direction that the user was required to shift to self-correct their AWD and stabilize their balance ratio. When participants' balance is stabilized the vibro-tactile notification stops, indicating a 50 : 50 balance has been achieved.

### 4.5.3 EMS feedback and Auto-correction:

Upon AWD detection, the EMS feedback is activated to apply the recorded EMS intensity to the tibialis muscles in the opposite leg to the AWD lean. This invokes an involuntary muscle contraction to produce a counter-weight shift force in the opposite direction to the AWD lean for stabilizing the balance. Figure 1(A) and (C) illustrate the AWD left and right-leaning posture, respectively. Figures 1(B) and (D) illustrate the automatically corrected posture after EMS has been applied. The EMS is deactivated when the balance ratio stabilization has been achieved.

## 5 RESULTS

The average number of AWD conditions observed per participant in the quiet standing application was (12.38, 13.05, and 14.11) for the audio, vibro-tactile, and EMS feedback modalities, respectively, and (12.22, 13.83, and 12.66) for the audio, vibro-tactile, and EMS feedback modalities, respectively in the mobile game application. For the quiet standing application, the mean EMS intensity required to correct AWD condition and stabilize balance posture was 50.55 $mA$ ($S.D = 9.05$ $mA$) while for the mobile game task, the mean EMS intensity was 51.94 $mA$ ($S.D = 8.25$ $mA$). To analyze the performance of our approach, we used repeated-measures 2-Factor *ANOVA* to determine the influence of modality and application types on each dependent variable and the consolidated results are presented in Table 2, 3, 4, 5. For the non-parametric user perception Likert scale data, we utilized the Aligned Rank Transform (ART) tool [110] and performed repeated measures 2-Factor *ANOVA* tests on the aligned ranks for the user perception Likert scale data.

### 5.1 Average Correction Response Times

For H1, the main effect for modality type yielded an $F(2,68) = 125.16$, $p < 0.001$, indicating a significant difference between Audio ($M = 2.58$, $S.D = 0.63$), Vibro-tactile ($M = 1.8$, $S.D = 0.45$), and EMS modalities ($M = 1.32$, $S.D = 0.29$) as illustrated in Figure 10 (a). A post-hoc pairwise comparison with Bonferroni correction conducted on the average correction response times across

Table 2: 2-Factor *ANOVA*: Average Correction response times (ACRT). M: Modality, A: Application.

| Source | ACRT | p |
|---|---|---|
| M | $F(2,68) = 125.16$ | $< 0.001^*$ |
| A | $F(1,34) = 2.744$ | $0.107$ |
| M X A | $F(2,68) = 5.803$ | $0.016^*$ |

*Note: * indicates significant difference $p < 0.05$.*

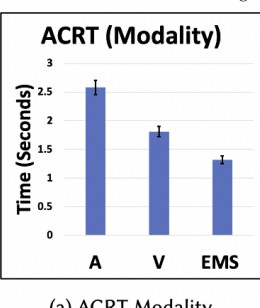
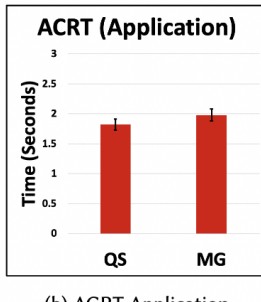

(a) ACRT Modality   (b) ACRT Application

Figure 10: Average correction response times (ACRT) across (a) Modality and (b) Application. Error bars:95% CI.

the three modalities showed that EMS feedback modality was significantly faster than the audio modality ($t_{34} = -1.262$, $p < 0.001$), and the vibro-tactile feedback modality ($t_{34} = -0.492$, $p < 0.001$). The main effect for application type yielded an $F(1,34) = 2.744$, $p > 0.05$, indicating that the effect of application type was not significant between quiet standing ($M = 1.8$, $S.D = 0.6$), and mobile game ($M = 2$, $S.D = 0.79$) as illustrated in Figure 10 (b). The interaction effect was significant $F(2,68) = 5.803$, $p < 0.05$. Significant differences were found in the system performance with regards to average correction response times between different feedback modalities with EMS feedback delivering the fastest correction. As a result, we were able to accept H1.

## 5.2 User Perception of Correction Feedback Accuracy

For H2, the main effect for modality type yielded an $F(2,68) = 4.113$, $p < 0.05$, indicating a significant difference between Audio ($M = 5.83$, $S.D = 1.03$), Vibro-tactile ($M = 6.44$, $S.D = 0.69$), and EMS modalities ($M = 6.67$, $S.D = 0.53$) as illustrated in Figure 11 (a). A post-hoc pairwise comparison with Bonferroni correction conducted on the participants ranking of correction feedback accuracy across the three modalities showed significant differences between the audio and vibro-tactile ($t_{34} = -0.611$, $p < 0.001$), and audio and EMS feedback types ($t_{34} = -0.833$, $p < 0.001$) but no evidence of significant differences between the vibro-tactile and EMS feedback. The participants perceived EMS feedback to be more accurate than the audio, but not vibro-tactile feedback and hence we were not able to accept H2. The main effect for application type yielded an $F(1,34) = 0.052$, $p > 0.05$, indicating that the effect of application type was not significant between quiet standing ($M = 6.3$, $S.D = 0.82$), and mobile game ($M = 6.33$, $S.D = 0.81$) as illustrated in Figure 11 (b). The interaction effect was not significant $F(2,68) = 2.988$, $p > 0.05$.

## 5.3 User Perception of Comfort

For H3, the main effect for modality type yielded an $F(2,68) = 1.376$, $p > 0.05$, indicating no significant difference between Audio ($M = 6.3$, $S.D = 0.98$), Vibro-tactile ($M = 6.36$, $S.D = 0.96$), and EMS modalities ($M = 5.91$, $S.D = 1.23$) as illustrated in Figure 12 (a). The main effect for application type yielded an $F(1,34) = 1.364$, $p > 0.05$, indicating that the effect of application type was not significant between quiet standing ($M = 6.43$, $S.D = 1.02$), and

Table 3: 2-Factor *ANOVA*: User Perception-Correction feedback accuracy (CFA). M: Modality, A: Application.

| Source | CFA | p |
|---|---|---|
| M | $F(2,68) = 4.113$ | $0.021^*$ |
| A | $F(1,34) = 0.052$ | $0.82$ |
| M X A | $F(2,68) = 2.988$ | $0.057$ |

*Note: * indicates significant difference $p < 0.05$.*

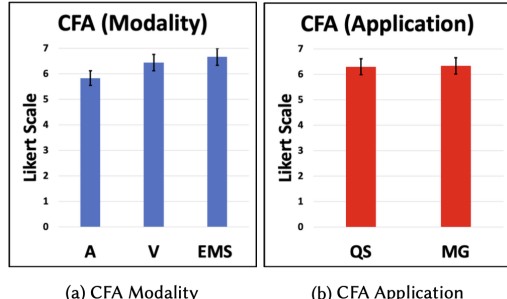

(a) CFA Modality   (b) CFA Application

Figure 11: User perception of correction feedback accuracy (CFA)across (a) Modality and (b) Application. Error bars: 95% CI.

mobile game ($M = 6$, $S.D = 1.08$) as illustrated in Figure 12 (b). The interaction effect was not significant $F(2,68) = 2.027$, $p > 0.05$. As no significant differences were found in the main effects for modality or the application type, neither modality nor application had any influence on the user comfort. As a result, we accept H3.

## 5.4 User Perception of Task Disruption

For H4, the main effect for modality type yielded an $F(2,68) = 0.036$, $p > 0.05$, indicating no significant difference between Audio ($M = 2$, $S.D = 1.37$), Vibro-tactile ($M = 2.11$, $S.D = 1.30$), and EMS modalities ($M = 2.28$, $S.D = 1.65$) as illustrated in Figure 13 (a). The main effect for application type yielded an $F(1,34) = 0.280$, $p > 0.05$, indicating that the effect of application type was not significant between quiet standing ($M = 1.7$, $S.D = 1.05$), and mobile game ($M = 2.51$, $S.D = 1.67$) as illustrated in Figure 13 (b). The interaction effect was not significant $F(2,68) = 1.427$, $p > 0.05$. As no significant differences were found in the main effects for modality or the application type, neither modality nor application had any influence on task disruption. As a result, we reject H4.

## 5.5 User Perception and Preference

Mean rankings for user perception of correction feedback accuracy, posture awareness, comfort, and task disruption are shown in Figure 14. Participants ranked their posture awareness on a 7-point scale where 1 means not at all aware and 7 means completely aware. Participants' ranking indicated higher posture awareness ($M = 5.46$, $S.D = 1.61$) in the quiet standing task, while posture awareness was significantly reduced for the mobile game condition ($M = 2.33$, $S.D = 1.27$). Additionally, when participants were asked about their preferred modality for correcting AWD, 55.56% of the study population reported that EMS feedback was their preferred correction feedback technique, while 36.11% preferred the vibro-tactile feedback and 8.33% preferred the audio feedback. However, 29 out of 36 participants reported that they would be willing to purchase EMS feedback for AWD posture correction if it were a commercially available product. Participants also ranked their shared responsibility with auto-correction utilizing EMS on a 7-point scale where 1 means not at all and 7 means completely. The mean shared responsibility exhibited by the participants was 2.00 ($S.D = 1.08$) in the quiet standing task, and 1.72 ($S.D = 0.75$) for mobile game condi-

Table 4: 2-Factor *ANOVA*: User perception-Comfort. M: Modality, A: Application.

| Source | Comfort | p |
|--------|---------|---|
| M | $F(2,68) = 1.376$ | 0.259 |
| A | $F(1,34) = 1.364$ | 0.251 |
| M X A | $F(2,68) = 2.027$ | 0.14 |

*Note: ∗ indicates significant difference $p < 0.05$.*

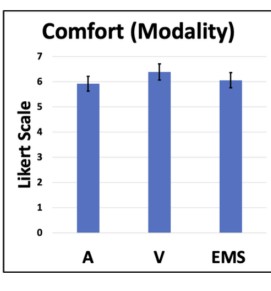
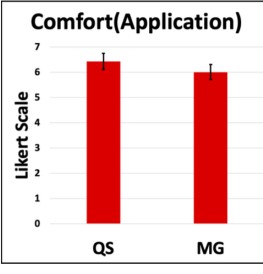

(a) Comfort Modality  (b) Comfort Application

Figure 12: User perception of comfort across (a) Modality and (b) Application. Error bars: 95% CI.

Table 5: 2-Factor *ANOVA*: User Perception-Task disruption (TD). M: Modality, A: Application.

| Source | TD | p |
|--------|----|---|
| M | $F(2,68) = 0.036$ | 0.965 |
| A | $F(1,34) = 0.280$ | 0.6 |
| M X A | $F(2,68) = 1.427$ | 0.247 |

*Note: ∗ indicates significant difference $p < 0.05$.*

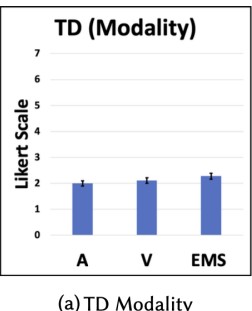
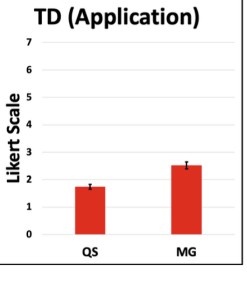

(a) TD Modality  (b) TD Application

Figure 13: User perception of task disruption (TD) across (a) Modality and (b) Application. Error bars: 95% CI.

tion. Participants ranked EMS feedback to be a highly interesting concept for automatic AWD correction with a mean ranking of 6.33 ($S.D = 1.39$) on a 7-point Likert scale.

## 6 DISCUSSION

Given the recent developments of EMS feedback in accelerating preemptive reflexes [51, 52, 84], and slouching posture correction [53], we were interested in understanding if EMS feedback could be utilized for correcting AWD. In comparison to the alternative techniques, we find there are several benefits to automatic correction using EMS. Our approach was able to achieve significantly faster correction at a high accuracy while delivering an equally comfortable user experience across different tasks with different levels of engagement and posture awareness. Although research on postural control, sway analysis, and AWD alert systems have been conducted, the system's correction responsiveness and user perception have not been measured or reported. Therefore, our study primarily focuses on evaluation of the performance and user perception of our EMS feedback based automatic AWD detection and correction technique against traditional audio and vibro-tactile feedback mechanisms.

Correction response times were measured from the time correction feedback is activated until balance has been restored. The average correction response times were significantly faster for the EMS feedback modality in comparison to the audio and vibro-tactile modalities. In both application types, the EMS modality delivered faster AWD corrections leading to faster stabilization and restoration of balance as illustrated in Figure 15. This was also reflected in the participants' comments on EMS: *"the fastest feedback and made me correct the best"*, *"liked the fast response"*, and *"Perfect response, subtle but noticeable"*. The faster correction response times to EMS feedback could be mainly due to the automatic stabilization and balance restoration which does not require the user to place emphasis on processing audio or vibro-tactile feedback prior to engaging in a self-assessment and self-correction process. This self-assessment and self-correction process in the audio and vibro-tactile feedback mechanisms place an additional cognitive load on the user while being engaged in their task and rely entirely on the user's willingness or intent to self-correct their posture. One participant's comment attests to this fact: *"Audio-took me time to process the feedback command and then correct, Vibration- got my attention, EMS-pulling quickly didn't need my attention"*. On the contrary, EMS feedback

which does not require the participants' attention in the correction process, thereby allowing one to continue leveraging the cognitive or attentional resources for the primary task which would have otherwise been required for auditory, visual or sensory processing for postural control. Results also indicate that application type had no effect on the correction response times suggesting that EMS would be capable of delivering faster correction responses across a range of applications with varying levels of engagement and posture awareness. This frees up the cognitive demand of the visual, vestibular, and proprioception placed on the user and makes it especially beneficial as a smart intervention technique for athletes in post-operative rehabilitation to prevent unnecessary AWD conditions that prohibit or impede recovery, mitigating risk of re-injury, rebuilding strength and motion, and restoring normal function thereby ensuring proper recovery and safer return-to-sport.

Participants' ranking of their perceived accuracy of correction feedback indicated that EMS feedback was more accurate than the audio, and equally accurate in comparison to the vibro-tactile feedback. Some of the participants' comments reflected this fact: *"Audio was most distracting"*, *"EMS was a better form of feedback, was strong and detected even the slightest imbalance"*, *"EMS gave me best feedback, I couldn't hear the audio feedback over the game"*, *"EMS most accurate and best for correction, but could be uncomfortable for some people"*. The participants perceived accuracy of EMS and vibro-tactile feedback equally well and this may have been due to the nature of explicit somatosensory confirmation provided by these two feedback types during delivery and termination of correction feedback when AWD is detected and corrected, respectively.

Participants' ranking their perceived level of comfort and task disruption, indicated neither modality nor application had any influence on the user comfort or task disruption. Although, both EMS and vibro-tactile feedback types are non-invasive in nature, EMS feedback has been known to produce a stronger somatosensory experience due to its ability to produce an involuntary muscular contraction along with a vibro-tactile effect. However, participants perceived all three modalities to be equally comfortable and equally disruptive. This could be due to careful calibration for an optimal EMS intensity that provides the user with a comfortable experience while generating a physiological response to effect a counter-weight shift. This user perception of comfort and task disruption illustrates participants' acceptance of EMS feedback as a viable alternative to

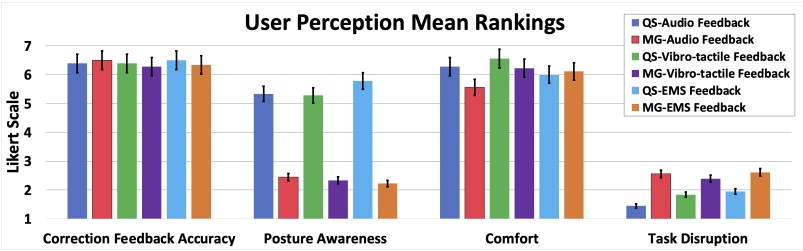

Figure 14: User perception mean rankings for correction feedback accuracy, posture awareness, comfort, and task disruption across all modality and application types. Likert Scale: 1-meaning not at all, 7-meaning completely. QS:Quiet Standing, MG:Mobile Gaming. Error bars: 95% CI.

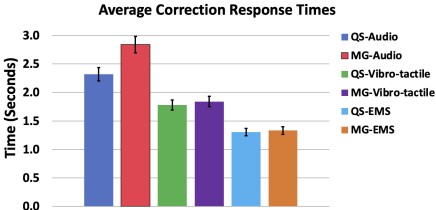

Figure 15: Average Correction Response times across all modality and application types. Error bars:95% CI.

the traditional feedback mechanisms with the additional advantage of automatic posture correction freeing up cognitive resources to focus on more important tasks. Participants comments show that EMS *"took time getting used to. It is like an Assisted PUSH, very useful when physical awareness is lacking"* and *"The pulling effect surprised me a bit but it was fine after"*. This acceptance shows EMS feedback's potential to be developed as a commercial product and allow EMS-based smart intervention wearable technology to be available for everyday use especially by younger adults engaging in the use of mobile devices for gaming, social media consumption while standing, and older adults engaging in work related activities in industrial, manufacturing or customer service sectors that require long standing hours. This fact was also supported by the participants' willingness (80.55% of healthy study population) to purchase EMS based wearable AWD intervention technology if it were available as a commercial product.

It was also interesting to note that the EMS intensity required for effecting counter-weight shift by stimulating the tibialis muscles was higher in comparison to another study on automatic detection and correction of slouching [53] where slouched posture was corrected by stimulating the trapezius muscles ($Mean\ EMS\ intensity$ : $Tibialis = 51.25\ mA, Rhomboid = 43.47\ mA$). This may be because the rhomboid muscle is more accessible physiologically in comparison to the tibialis muscle which is regarded as more deeper muscle group and thereby necessitating higher EMS intensity to recruit the motor neurons to cause an involuntary muscular contraction and generate a physiological response for producing the counter-weight shift effect with the desired magnitude and in the desired direction. Participants also reported shared responsibilities in helping/aiding the correction process during the EMS feedback session. This illustrates the participants' adaptability to new technology and demonstrates the positive learning effect produced by the EMS feedback towards better postural control. Further, it also demonstrates that EMS feedback with its somatosensory feedback encouraged the participants to get involved in the correction process. Finally, one participant commented *"It's like trainer wheels on a bicycle"*, while some participants commented that EMS *"Felt amazing"*, *"Auto-correction is good"*, *"the fastest feedback and made me correct the best"*, and *"correction happens without thinking about it"*.

Finally, our system could be particularly beneficial in preventive health care and the development of rehabilitation protocols for recovery post-knee/ankle surgery as it would allow the healthcare specialists to develop customized recovery protocols for different individuals by varying the balance and time thresholds, and EMS intensity parameters as prescribed. This would ensure precision control of the weight distribution on the operated leg at different stages of recovery to maximize rebuilding strength and mobility, and minimizing the time duration for return-to-sport in case of athletes or return-to-normal function in case of non-athlete patients. Also, our EMS feedback when integrated with load sensors and IMUs embedded in shoes, could be utilized to detect AWD and dangerous tilt angles for automatic fall prevention in older adults, and PD patients who present a higher risk of injury due to falls experienced through the loss of balance. Therefore, our autonomous AWD detection and correction system could be a useful alternative or inclusion to existing environment, health, and safety (EHS) guidelines for mitigating risk of workplace injury, improving employee health, and in rehabilitation and preventive health care.

## 7 LIMITATIONS AND FURTHER WORK

One prominent limitation is the need to manually place electrodes on the body. To resolve this, we plan to integrate the electrodes into wearable clothing and devising an auto-calibration system that can be customized to each individual's comfort. Another limitation of our study is that although our system detects any imbalance instantly, we utilized a time threshold of 2.9$s$ to discriminate AWD conditions from other actions. However, this threshold could be shortened if our AWD detection system were integrated with IMU sensors to classify non-AWD actions. Our future work includes the development of a mobile application to allow participants to customize the balance ratio, time thresholds, and EMS intensity. We also plan to gather data on how people with impaired balance issues fall compared to a healthy person's fall and implement an automatic fall prediction and prevention system utilizing EMS.

## 8 CONCLUSION

We have demonstrated that our automatic EMS-based physiological feedback loop is a viable approach to supporting AWD detection and correction, and stabilizing balance through a counter-weight shift approach. Our auto-correction system utilizing EMS feedback demonstrated significantly faster posture correction response times compared to the self-correction in the audio and vibro-tactile feedback. Our approach also showed that participants perceived EMS feedback to be highly accurate, equally comfortable, and produced no more disruption than the alternative techniques it was tested against in both the quiet standing and the mobile game applications even though the posture awareness across the application types were significantly different. Therefore, automatic AWD detection and correction utilizing EMS shows promising results and can be developed as an alternative method for AWD correction.

### ACKNOWLEDGMENTS

This work is supported in part by NSF Award IIS-1917728. We also thank the anonymous reviewers for their insightful feedback.

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
