# OpenReview forum: "Automatic Asymmetric Weight Distribution Detection and Correction Utilizing Electrical Muscle Stimulation"
_graphicsinterface.org/Graphics_Interface/2022/Conference — GI 2022_

### Official Review · Reviewer_GGGG · 2022-01-06
**Automatic Asymmetric Weight Distribution Detection and Correction Utilizing Electrical Muscle Stimulation**

**Rating:** 7
**Confidence:** 2

**Review:**

This submission presents a controlled experiment examining EMS as a feedback mechanism to help users correct posture imbalances while standing.  Results show advantages of using EMS as compared to visual or tactile approaches.

Strengths

This submission is a good fit for Graphics Interface.  It describes a small contribution that might not be sufficient for a top-tier venue, yet, to my knowledge as an outsider on the topic, contributes sufficient new knowledge that the community would benefit from its dissemination.

The submission does a great job of overviewing related work in both the application domain and contextualizing the work within the HCI literature.

The study is clearly described, appears to have been well executed and has produced some significant results that others can hopefully build on (see a few small caveats below).

Weaknesses

There are a couple of issues with the analysis that I believe could be fixed in a camera-ready version without changing the paper’s overall story.

- The paper should specify which corrections were applied for the pairwise comparisons (e.g., Bonferroni, Tukey)
- Only main effect analysis is report for H1, which Is not sufficient to accept H1.  Based on the graph, I am confident that the post-hoc comparisons will be significant, they just have to be included.
- Finding a non-significant result is not the same as equivalence testing and therefore H3 should not be accepted based on the analysis presented.  For example, a result might not reach significant due to lack of power. One strategy is to instead list this as an expectation rather than a hypothesis to be accepted or rejected.
- The paper describes the data in Table 1 as “qualitative”. Given that these are in fact numbers,  perhaps the authors meant to refer to this data as “subjective”?

The discussion starts to go beyond the data, claiming to have demonstrated effects that have not yet been systematically or comprehensively tested.  This is particularly true at the start of the second column on p 10.  For example, given the short, controlled nature of the study, the study doesn’t fully demonstrate improved posture control or that participants are more involved in the process.  I think it is fine to state that there are promising initial results if the language is more cautious.  The controlled, short-term nature of the study should also be discussed as a limitation.

---

### Official Review · Reviewer_RuVZ · 2022-01-13
**The study is important and well designed and conducted. The results are clear as well and show promise in helping to correct AWD.**

**Rating:** 8
**Confidence:** 3

**Review:**

This paper reports the results of an experimental study that compared the response time and usability of three different techniques for correcting asymmetric weight distribution (AWD). The authors of the paper developed a technique that used EMS feedback with auto AWD correction, which was compared against two self-correcting techniques using audio feedback and vibro-tactile feedback respectively. The study was well-designed and reported with great details. The results showed that the auto-correction technique was well-perceived and promising that it could potentially help people correct their AWD which has been found to be a common issue.

A few comments for consideration:
•	The device used for producing the EMS, TENS, is better described early on.
•	What are the participants’ health status (other than being able-bodied), e.g., any neurological problems like numbness, or muscular issues?
•	Any gender/height/weight differences found?
•	What happens if the EMS feedback overshoots such that the user becomes unstable and possibly fall. This is important especially because the authors suggested that the technique may be used to detect and prevent fall in older adults.

---

### Official Review · Reviewer_gToi · 2022-01-14
**Good contribution overall, pairwise comparison not presented or not done (with it paper would be "clear accept"), some limitations not discussed**

**Rating:** 7
**Confidence:** 5

**Review:**

The manuscript presents an electrical muscle stimulation system that automatically stimulates lower limb muscles in order to automatically restore symmetrical weight distribution (a center of pressure that is centralized, measure using a wii balance board). The device is presented as is a user study that compares it to two existing systems: audio and haptic feedback. They are compared in two conditions, while standing still and while playing a mobile game.

The manuscript is generally well written although it is a little bit long. The introduction and related work sections, in particular, could be shortened and tightened up.

The methodology is clearly presented and most important details are given. The study has a good number of participants for a conference paper (36). The results are interesting and represent a contribution to existing knowledge. However, the discussion and conclusion oversell these results a little and fail to discuss some of the limitations, in my opinion.



Major issues:


In the methodology, no method is given for the post-hoc pairwise comparisons

The results for correction response time only show that the modality had an effect. No pairwise comparison is presented. This therefore does not show that the time with EMS is statistically lower than the time for haptic feedback, which is probably THE main finding of the study!

The methods that are compared to EMS are methods that require the user to consciously perceive the feedback and adjust his balance. With such methods, users would typically become more efficient with learning. This would not necessarily but true of the EMS methods which auto-corrects. Therefore, comparing the methods with untrained users may bias the results towards the automatic method (EMS).

In the discussion, the authors claim that EMS imposes less of a cognitive load than haptic or auditory feedback. While this makes intuitive sense, it is not supported by their results. Indeed, their results show no difference between the standing still and gaming conditions. Had EMS showed a greater advantage in the gaming mode vs standing still mode, this would be supported.




Minor issues and questions requiring more details:


The stimulation causes a movement of inversion (foot roll). This would not be the typical way one would correct a balance asymmetry. What is the impact of correcting like this? Although this surpasses the scope of the paper, it should be discussed given that it is presented as a clinical tool. For example, foot inversion would affect knee varus and knee moments.

How were the specific muscles chosen?

Were the electrodes placed by a clinician? By someone who had experience using EMS?

EMS intensity is chosen manually, so it is different for every participant in the study. Moreover, the threshold of activation is confirmed visually, which is probably not very repeatable.

I would have liked to see a figure showing the trajectory of the COP for EMS, vs other more intentional corrections. The EMS could act as a perturbation with a larger damping period, caused by the users reaction to this perturbation?

---

### Decision · Program_Chairs · 2022-01-18

Accept